# Effect of Dancing Interventions on Depression and Anxiety Symptoms in Older Adults: A Systematic Review and Meta-Analysis

**DOI:** 10.3390/bs14010043

**Published:** 2024-01-10

**Authors:** Tiago Paiva Prudente, Eleazar Mezaiko, Erika Aparecida Silveira, Túlio Eduardo Nogueira

**Affiliations:** 1Faculty of Medicine, Federal University of Goiás (UFG), Goiânia 74605-050, Brazil; tiagopaiva@discente.ufg.br (T.P.P.); erikasil@terra.com.br (E.A.S.); 2Faculty of Dentistry, Federal University of Goiás (UFG), Goiânia 74605-020, Brazil; eleazarmezaiko@discente.ufg.br

**Keywords:** aged, aging, anxiety, depression, dancing, cognitive impairment, physical exercise, psychology, quality of life

## Abstract

This systematic review and meta-analysis (PROSPERO CRD42023428105) investigated the effect of dancing on depression and anxiety symptoms in older adults. Conducted up to October 2023, the search across seven databases and gray literature yielded 5020 records. Only randomized trials that analyzed dance interventions for depression and/or anxiety in older adults were included. Nineteen randomized trials, involving 508 participants in dance classes lasting 5 weeks to 18 months, were included and 16 were subjected to meta-analysis. Risk of bias was assessed using the Cochrane tool. The meta-analysis showed a statistically significant reduction in depression among older adults participating in dance interventions (*p* < 0.01). A decrease in depressive symptoms was significant compared to that in those involved in no other intervention (*p* = 0.02) but not compared to that achieved with other interventions in control groups (*p* = 0.96). Subgroup analysis showed no significant differences in depression scores for those with mild cognitive impairment (*p* = 0.47). These conclusions are associated with moderate bias and very low certainty. Due to heterogeneity and the small number of studies, conclusions for anxiety outcomes could not be drawn. These results underscore the potential clinical relevance of integrating dance into mental health interventions for older adults, thereby highlighting a promising avenue for enhancing the mental well-being of this demographic.

## 1. Introduction

The global population of individuals aged 60 and above is steadily increasing and reached approximately one billion in 2019 [1]. Psychiatric disorders within this population significantly contribute to the burden of disease and disability [1]. Depression is one of the most prevalent geriatric psychiatric disorders, affecting approximately 32% of this population worldwide [2]. Anxiety, another prominent mental health issue, frequently coexists with depression, afflicting up to 50% of the older population [3].

Nonpharmacological approaches are the first line of treatment for such conditions, especially for mild cases, and may involve the life review process, problem-solving therapy, and cognitive-behavioral therapy [4]. A recent systematic review (SR) and meta-analysis showed that physical activity (aerobic and resistance exercise) effectively treats depressive symptoms in adults, mainly when performed under supervision [5]. However, the effectiveness of some types of physical activity, such as dancing, at controlling anxiety and depression has been little investigated in the older population.

Dance is defined as a successive series of rhythmic steps or body movements performed in coordination with music [6]. Studies have highlighted the positive impact of dance on both physical and mental health, showcasing significant improvements in fitness levels, mood, social interactions, and overall quality of life through a comprehensive blend of objective and subjective evaluations [7,8,9]. Prior randomized controlled trials and SR focusing on younger populations have consistently shown that dancing effectively alleviates symptoms of depression and anxiety [10,11]. Moreover, primary studies involving older adults have shown promising results, highlighting the positive impact of dancing in reducing symptoms of depression and anxiety within this population. Various dance styles have been investigated, including square dance [12,13], poco-poco [14], jazz [15], and traditional Greek dance [16]. Furthermore, researchers have explored diverse contexts, such as long-stay institutions for older adults [17] and among individuals with and without cognitive impairment [14,18].

Despite existing research on the effects of dance among the older population, SRs encompassing various dance types and diverse conditions remain necessary. Depression in older adults with cognitive impairment and Parkinson’s disease (PD), for example, have remarkable prevalence rates of 32% [19] and 30% [20], respectively. Further investigation is required to explore the potential influence of these noteworthy conditions on the effects of dance in older adults with depression. One SR specifically focused on older adults with cognitive impairment found no significant improvement in the assessed scores for depression [21]. Another SR evaluated those diagnosed with PD and, through a qualitative synthesis of the literature, concluded that dance held the potential to improve mood [22].

Two additional recent SRs with meta-analysis on the topic have also been published. Nevertheless, one of them conducted searches only on three databases, analyzing broader aspects of psychological health [23], while the other did not perform a comprehensive subgroup analysis, combining studies with diverse methods, as well as subjects receiving different interventions and with different comorbidities [24]. In this sense, some authors demanded further studies with more rigorous methods to investigate the effect of dance interventions on mental health outcomes such as depression [25,26].

The World Health Organization (WHO), in collaboration with the United Nations (UN), has designated the decade 2020–2030 as the United Nations Decade of Healthy Aging. This initiative calls upon all sectors of society to prepare for healthy aging by improving skills, competencies, and knowledge [27]. Therefore, in addition to the previously mentioned justifications, this review aligns with the goals set forth by the WHO and the UN. It has the potential to generate valuable evidence that can be employed to enhance the quality of life of older adults. Thus, this SR and meta-analysis aimed to answer the following research question: What is the effect of dance on symptoms of depression and anxiety in older adults?

## 2. Materials and Methods

This systematic review and meta-analysis adhered to the PRISMA guidelines in its conduct and reporting [28]. The study protocol was registered in the PROSPERO database (registration number CRD42023428105). The research question was formulated using the PICOS acronym, in which:P (Population): Older adults (as defined by each study or, in cases where the population was not specifically restricted, study sample with a mean age of 60 or older [29]).I (Intervention): Dance.C (Comparator): Other types of intervention or no intervention, as well as participants’ comorbidities.O (Outcome): Symptoms of depression and/or anxiety.S (Study design): Randomized controlled clinical trials.

### 2.1. Search Strategy and Study Selection

The search strategy was initially developed for PubMed using combinations of controlled terms, keywords, and Boolean operators. An expert librarian was consulted to refine and tailor this strategy for additional databases, including EMBASE, Scopus, LILACS, Web of Science, Livivo, PsycINFO, and SportsDiscus. The searches were conducted in March 2023, and alerts for new publications were created for each database. The search was updated in October 2023 (Appendix A). Additionally, gray literature sources such as Google Scholar and Proquest were considered, limited to the first 100 results [30].

A manual search was also carried out in the references of the included publications. There was no restrictions regarding publication date. The retrieved publications were exported to EndNote (EndNote version 21, The EndNote Team, Philadelphia, PA, USA), where duplicates were automatically removed. Subsequently, records were exported to Rayyan (Rayyan’s web software, Qatar Computing Research Institute, Doha, Qatar) for manual exclusion of duplicates and to conduct the screening phase, which was performed independently by two calibrated evaluators (TPP and EMVD).

The first phase of the selection process involved screening titles and abstracts, followed by phase two, during which a thorough examination of the screened publications was carried out using the full-text records. Conflicting decisions encountered during both phases were resolved through discussion with a third reviewer (TEN) to reach a consensus. In situations where accessing full-text publications was not feasible, attempts were made to obtain the full text by contacting authors via email and utilizing specialized library services. In case of unsuccessful attempts, the respective publications were subsequently excluded.

### 2.2. Eligibility Criteria

Studies were considered eligible if they were randomized clinical trials, written in the Roman alphabet, included older adults diagnosed with depression and/or anxiety, applied any form of dance as a therapeutic and/or supportive intervention, comparing the data with a control group in which no dance intervention or other interventions were used; and evaluated changes in symptoms of depression and/or anxiety using validated psychometric instruments.

Publications were excluded if the participants were older adults with medical conditions that hindered their ability to follow instructions and if they had multiple psychiatric conditions (e.g., dementia, schizophrenia, schizoaffective disorder, etc.). Studies combining dance with other activities, such as meditation and supervised walking, were also excluded.

### 2.3. Data Extraction

The data from the studies that met the eligibility criteria were independently extracted by two authors (TPP and EMVD) onto a standardized sheet, which contained the following information: study characterization (authors, year of publication, and country); population characteristics (sample size and sociodemographic details such as gender, age, and comorbidities); intervention characteristics (type of dance, duration, and periodicity); measures of anxiety and depression symptoms before and after the intervention; and other relevant results, including effect measures (mean difference, odds ratio, correlation coefficient, or other reported measures).

### 2.4. Risk of Bias

The risk of bias in the included studies was assessed using the Cochrane risk-of-bias tool for randomized trials (RoB2). This tool comprises five domains: bias arising from the randomization process, bias due to deviations from intended interventions, bias in the measurement of outcomes, bias pertaining to the selection of the reported result, and bias concerning missing outcome data. The risk of bias for each domain, as well as the overall risk of bias, were judged following the recommendations provided by the tool [31]. The studies considered to have a high risk of bias were not included in the meta-analysis.

### 2.5. Publication Bias

Publication bias was appraised using the rank correlation test [32], the fail-safe N method [33], and the regression test for funnel plot asymmetry [34].

### 2.6. Certainty of Evidence

The certainty of evidence was assessed using the Grades of Recommendation, Assessment, Development, and Evaluation (GRADE) system. This tool evaluates five domains: risk of bias, inconsistency, indirectness, imprecision, and publication bias. The certainty level for the body of evidence is categorized as high, moderate, low, or very low [35].

### 2.7. Statistical Analysis

The software Review Manager (version 5.3) was used for the meta-analysis [36]. Data from included studies were combined using the random-effects technique due to variations in psychometric scales used across studies. Heterogeneity among studies was calculated using Cochran’s Q test and the I^2^ index. Studies were considered as having insignificant heterogeneity if the I^2^ index was 0–30%, moderate heterogeneity for 30–50%, substantial heterogeneity for 50–90%, and considerable heterogeneity for above 90% [37]. Subsequently, standardized mean differences in depression/anxiety scores were calculated with and without dance interventions, along with their corresponding confidence intervals. The pooled analyses were based on the DerSimonian and Laird random-effects model and inverse-variance method [38].

A subgroup analysis was also conducted to assess whether comorbidities influenced the meta-analysis results. Finally, publication bias was assessed with statistical software Jamovi (version 2.3) [39] using an Egger funnel plot and statistical tests. All statistical analyses were performed at a predetermined significance level of 0.05.

## 3. Results

### 3.1. Selection Process

The systematic search retrieved 5020 records. Following the elimination of duplicate entries, the titles and abstracts of 3737 articles were screened, resulting in the exclusion of 3675 records. Among the 62 full-text records sought, 5 could not be retrieved. Subsequently, 57 full-text publications were assessed for eligibility, and 16 were later included. Additionally, a manual search of reference lists of the included publications led to the inclusion of three additional publications. Thus, a total of 19 publications were included in this systematic review, of which 16 underwent meta-analysis. A detailed overview of the evidence source selection is presented in Figure 1.

### 3.2. Study Characteristics

#### 3.2.1. Anxiety

The characteristics of the included studies are summarized in Table 1. Of the five publications that analyzed anxiety [16,40,41,42,43], a total of 319 individuals were included (180 in the intervention groups and 139 in control groups). The majority of participants were women (approximately 65%), with overall mean age ranging from 64.1 to 77.3 years. None of the included studies provided information on whether participants were using medications to control anxiety symptoms.

The studies were carried out in Brazil [42], China [43], Greece [16], and Spain [40,41]. The intervention groups were engaged in dance sessions with a mean of 150 min/week for three months. The dance styles incorporated in the studies were diverse, including ballroom dance [42], aerobic dance [41], Greek traditional dance [16], dance therapy [40], and square dance [43]. Two of the included studies analyzed patients with specific comorbidities, namely, mild cognitive impairment (MCI) [41] and hypertension [40].

In two studies, control groups encompassed individuals who continued their usual activities, without receiving any intervention [40,42]. The other three studies included groups receiving other types of intervention for comparison with the intervention (dance) groups, such as physical therapy [41], talk therapy [16], and other types of exercises (i.e., swimming, running, and tai chi) [43].

The tools used to evaluate anxiety in the studies were also diverse, including the Beck Anxiety Inventory (BAI) [42], the Hospital Anxiety and Depression Scale (HADS) [41], the State-Trait Anxiety Inventory (SAI) [16], the European Quality of Life 5 Dimensions (EuroQoL-5D) [40], and the Hamilton Anxiety Scale (HAMA) [43].

The follow-up duration of the studies varied, with some spanning 2 [16,40] to as long as 18 months [43]. Additionally, a few studies involved dancing classes conducted over three [41] and four months [42].

#### 3.2.2. Depression

For the 17 studies evaluating depression [17,40,41,43,44,45,46,47,48,49,50,51,52,53,54,55,56], a total of 870 individual were analyzed, with 397 participants receiving dancing interventions and 473 serving as controls. The majority of participants were women (approximately 74%), with overall mean age ranging from 64.1 to 84.0 years. During the study period of the included publications, only 36 participants were documented as using antidepressant medications: 29 participants in one study [17], 5 in another [49], and 2 in the last [45]. One publication explicitly mentioned that none of the participants were under any medication, including antidepressants [16]. This information was not reported in any of the other included studies.

Of the included studies, four were carried out in China [43,47,54,55], three in the USA [49,51,56], two in Spain [40,41], two in Brazil [45,46], and one in each of the following countries: Canada [52], Czech Republic [17], France [44], Italy [53], South Korea [50], and Turkey [48].

The dance classes were carried out over a mean duration of 120 min/week for four months. Specifically, four studies included individuals with PD [50,51,52,53], five studies exclusively enrolled only those with MCI [41,44,47,55,56], and one study focused on participants with hypertension [40].

The dance styles encompassed a broad range, including Latin dance [56], square dance [43,47,54], improvised dance [44], tango [52], ballroom dance [49], dance therapy [40,51], aerobic dance [41,55], belly dance [45], folkloric dance [48], Sardinian folk dance [53], video-game-based dance [50], or a combination of dance styles [17,46].

Regarding the characteristics of the control groups, participants either continued their routine activities [17,40,43,45,46,47,48,49,52,53,54,56] or underwent various interventions, such as talk therapy [51], physical therapy (i.e., training of motor abilities, such as strength, balance, and coordination) [41], neurodevelopment treatment [50], physician counseling [55], other types of exercises (i.e., swimming, running, Tai Chi, and others) [43,46], or aerobic exercises [44], all in accordance with the schedules followed by the dance-group counterparts.

Similar to anxiety, depression was assessed using a variety of instruments. The Geriatric Depression Scale (GDS) was employed in nine studies [17,44,45,47,48,49,54,55,56], the Beck Depression Inventory (BDI) in five studies [46,50,51,52,53], the Hamilton Rating Scale for Depression (HRSD) [49], the Hamilton Depression Scale (HAMD) [43], the EuroQoL-5D [40], and the HADS [41], each in one study.

Regarding the duration of follow-up, the studies spanned from 5 weeks [45] to 18 months [43], with the majority of them involving classes conducted over 3 to 4 months [17,41,44,52,53,54,55,56]. Notably, two studies included follow-up assessments conducted several months after the conclusion of the dancing intervention [49,55].

**Table 1 behavsci-14-00043-t001:** Characteristics of the studies included in this systematic review.

Author, Year	Country	Control Group (n)	Dance Group (n)	Comorbidities	Dance Type	Dance Schedule	Dance Program Duration	Control	Setting	Measurement of Depression *	Measurement of Anxiety *	Result (Intergroup)
			Anxiety									
Mavrovouniotis, et al., 2010 [16]	Greece	35	76	None	Greek traditional dance	60 min/session	2.5 months	Socialization	Health center	NA	SAI	↓
Alves, et al., 2013 [42]	Brazil	25	25	None	Ballroom dance	120 min 2 times/week	4 months	None	NR	NA	BAI	↓
			Depression									
Haboush, et al., 2006 [49]	USA	12	10	None	Ballroom dance	45 min 1 day/week	2 months	None	NR	GDS and HRSD	NA	—
Castro, et al., 2009 [46]	Brazil	20	20	None	Various	50 min 3 times/week	6 months	None	NR	BDI	NA	↓
Eyigor, et al., 2009 [48]	Turkey	18	19	None	Folkloric dance	60 min 3 times/week	2 months	None	Rehabilitation unit	GDS	NA	—
Vankova, et al., 2014 [17]	Czech Republic	83	79	None	Various	60 min 1 day/week	3 months	None	Nursing homes	GDS	NA	↓
Broadbent, et al., 2015 [45]	Brazil	7	7	None	Belly dance	45 min 2 times/week	5 weeks	None	Community center	GDS	NA	↓
Lee, et al., 2015 [50]	South Korea	10	10	PD	Video game dancing	30 min 5 days/week	1.5 month	NDT and FES	NR	BDI	NA	↓
Rios, et al., 2015 [52]	Canada	15	18	PD	Tango	60 min 2 days/week	3 months	None	NR	BDI	NA	—
Aguinaga, et al., 2016 [56]	USA	11	10	MCI	Latin dance	60 min 2 times/week	4 months	None	Day care center	GDS	NA	↑
Michels, et al., 2018 [51]	USA	4	9	PD	Dance therapy	60 min 1 day/week	2.5 months	Talk therapy	Movement studio	BDI	NA	NR
Zhu, et al., 2018 [55]	China	31	27	MCI	Aerobic dance	35 min 3 days/week	3 months	Physician counseling	Hospital	GDS	NA	—
Solla, et al., 2019 [53]	Italy	9	10	PD	Sardinian Folk dance	90 min 2 days/week	3 months	None	NR	BDI	NA	↓
Chang, et al., 2021 [47]	China	47	62	MCI	Square dance	30 min 3 times/week	4.5 months	None	Nursing homes	GDS	NA	↓
Ayari, et al., 2023 [44]	France	12	11	MCI	Improvised dance	60 min 1 day/week	4 months	Aerobic exercise	Day care center	GDS	NA	—
Zhang, et al., 2023 [54]	China	36	36	None	Square dance	60 min 4 days/week	4 months	None	Outdoors	GDS	NA	↓
			Anxiety and depression									
Zhang, et al., 2014 [43]	China	30	30	None	Square dance	30–60 min ≥4 days/week	18 months	Various	NR	HAMD	HAMA	↓ both
Serrano-Guzman, et al., 2016 [40]	Spain	35	32	HTN	Dance therapy	50 min 3 days/week	2 months	None	NR	EuroQoL-5D	EuroQoL-5D	↓ both
Bisbe, et al., 2020 [41]	Spain	14	17	MCI	Aerobic dance	60 min 2 days/week	3 months	Physical therapy	Hospital	HADS	HADS	— both

↑: Significantly increased; ↓: significantly decreased; —: no significant difference; BAI: Beck Anxiety Inventory; BDI: Beck Depression Inventory; EuroQoL-5D: European Quality of Life 5 Dimensions; FES: Functional electrical stimulation; GDS: Geriatric Depression Scale; HADS, Hospital Anxiety and Depression Scale; HAMA: Hamilton Anxiety Scale; HAMD: Hamilton Depression Scale; HRSD: Hamilton Rating Scale for Depression; HTN: hypertension; MCI, mild cognitive impairment; NA: not applied; NDT: neurodevelopment treatment; NR: not reported; PD: Parkinson’s disease; SAI: State-Trait Anxiety Inventory; * data measured as mean and standard deviation of the depression/anxiety scores across all studies.

### 3.3. Statistical Analysis

Out of the 19 included studies, 1 was not considered for the meta-analysis [48] due to the unavailability of data for the outcome of interest. Additionally, two studies were not included in the meta-analysis due to their high risk of bias [43,52]. Furthermore, one study did not provide results that allowed the interpretation of whether significant differences were detected [51].

#### 3.3.1. Anxiety

As a result, four studies showed a significant reduction in anxiety symptoms [16,40,42,43] among older adults following dance interventions. In summary, the meta-analysis, including the four studies, indicated a significant reduction in anxiety among older adults following dance interventions (SMD = −1.81; 95% CI: −3.59 to −0.04; *p* = 0.05), accompanied by considerable heterogeneity (I^2^ = 96%) (Figure 2A). Sensitivity analysis excluding studies that performed other types of intervention in the control groups did not change the significance.

#### 3.3.2. Depression

Among the studies investigating depression, nine indicated a decrease in depression symptoms [17,40,43,45,46,47,50,53,54] in older adults after dance interventions. Conversely, one study reported a significant increase in depressive symptoms [56], and six studies [12,41,44,48,49,52] did not indicate significant changes between the control and intervention groups following the dance interventions.

The meta-analysis, including 14 studies, showed that an overall statistically significant reduction in depressive symptoms was observed among older adults following dance interventions (SMD = −0.65; 95% CI: −1.12 to −0.17; *p* < 0.01) (Figure 2B). However, the subgroup analysis revealed that only participants without comorbidities exhibited a significant reduction in depression symptoms (SMD = −1.05; 95% CI: −1.95 to −0.16; *p* = 0.02), whereas those with PD (SMD = −0.73; 95% CI: −2.31 to 0.85; *p* = 0.37) and MCI (SMD = −0.15; 95% CI: −0.55 to 0.25; *p* = 0.47) did not show statistically significant differences compared to the control groups (Figure 2B).

A sensitivity analysis included eight studies that compared dance interventions with no intervention and continued to reveal a significant reduction in depression symptoms (SMD = −0.78; 95% CI: −1.45 to −0.11; *p* = 0.02), which was characterized by substantial heterogeneity (I^2^ = 91%) (Figure 2C). Subgroup analysis, accounting for comorbidities, also demonstrated a significant reduction in depression symptoms among older individuals without comorbidities (SMD = −1.05; 95%CI: −1.95 to −0.16; *p* = 0.02), while those with MCI did not exhibit significant changes (SMD = 0.02; 95%CI: −1.14 to 1.17; *p* = 0.98) (Figure 2C). Subgroup analysis for PD in studies with noninterventional comparison groups was not feasible due to the limited availability of only one study [53].

Furthermore, another sensitivity analysis included six studies that compared dance interventions with other interventions (i.e., aerobic exercises, talk therapy, physician counseling, and others). No significant difference in depression scores was observed (SMD = 0.01; 95% CI: −0.47 to 0.49; *p* = 0.96), and considerable heterogeneity was evident (I^2^ = 64%) (Figure 2D).

### 3.4. Risk of Bias

An overall moderate risk of bias was observed for the included studies (Figure 3). For the two studies solely investigating anxiety [16,42], this was primarily attributed to the first domain, as one of them lacked comprehensive details concerning the randomization process [16]. Additionally, neither of them provided information pertaining assessor blinding [16,42].

As for the studies solely investigating depression [17,44,45,46,47,48,50,51,52,53,54,55,56], seven of them did not explain how randomization was performed [17,44,46,48,49,50,52], and five did not mention assessor blinding [46,48,50,51,56]. One study explicitly stated that blinding was not implemented, resulting in a high-risk rating as well [52]. Lastly, three studies lacked information regarding missing outcome data [48,52,53].

The three studies investigating both anxiety and depression also presented bias in some domains. One of these studies did not explicitly mention randomization, leading to a high-risk assessment in this domain [43], while another did not present information on the randomization process [40]. Additionally, one of them was not clear at explaining the missing outcome data [43], while two did not report assessor blinding [40,43].

### 3.5. Publication Bias

Significant indications of potential publication bias were observed within anxiety studies, as corroborated by both the fail-safe N and Egger’s tests (*p* < 0.01; Figure 4A). Regarding studies examining depression symptoms, only the fail-N test indicated potential publication bias (*p* < 0.01), while rank correlation and Egger’s regression did not detect any significant asymmetry (*p* = 0.776 and *p* = 0.475, respectively; Figure 4B).

### 3.6. Certainty of Evidence

The certainty of evidence was appraised as very low both for depression and anxiety (Appendix A). This determination was primarily attributed to the heterogeneity in study methodologies, disparities in confidence intervals among studies, concerns related to indirectness, and instances in which the primary focus of certain studies was not to evaluate depression and/or anxiety. Notably, in the context of anxiety, the certainty of evidence should be regarded even lower than that of depression. This distinction arises from the identification of publication bias, as indicated by the funnel plot, and the inclusion of a smaller number of studies.

## 4. Discussion

Our systematic review and meta-analysis stands as the first in assessing the effect of dance compared to alternative interventions for the treatment of depression symptoms. Moreover, it contributes to the existing evidence on the effect of dance compared with non-interventional approaches for depression and anxiety symptoms. Our findings revealed a significant reduction in depressive symptoms among older adults who participated in dance interventions in comparison to those who did not, despite an overall moderate risk of bias. Furthermore, the meta-analysis showed that dance did not exhibit superior effectiveness compared to other interventions, many of which incorporated socialization elements. Regarding anxiety, although the results demonstrated a significant reduction in symptoms following dance interventions, we could not draw conclusions for this outcome due to the small number of studies and high heterogeneity. The study methods for both outcomes were considerably diverse, particularly regarding the type of dance interventions, the frequency of dance sessions, and the presence of comorbidities among the study participants.

Dancing is a versatile tool with the potential to serve as an effective intervention for addressing a wide range of both physical and psychological conditions across diverse populations [57,58,59]. One such condition is depression, which can result in various consequences, including an increased risk of cognitive decline, impaired executive functioning [60], frailty [61], and suicide and suicidal ideation [62]. Improvement in this outcome may largely depend on the duration and continuity of dancing interventions. In the included studies, interventions were carried out for a mean of approximately two classes of 60 min per week for 16 weeks. Such parameters generated significant reductions in depressive symptoms in almost 63% of the studies. However, this positive effect was not observed when participants were evaluated months after the studies concluded, implying the necessity of sustained dancing practices to improve depressive symptoms.

Conversely, six studies reported no significant difference in depression between the control and intervention groups [41,44,48,49,52,55]. This discrepancy could be attributed to the fact that, in these studies, fewer participants presented baseline depression. Additionally, in one of these studies, dance sessions were less frequent and shorter in duration compared to those in the other studies [49]. In addition, despite the absence of significant findings, one of the studies showed that participants self-reported feeling happier after dance interventions [48]. Surprisingly, one publication showed increased depressive scores in the intervention group compared to the control group [56]: it was a pilot randomized trial whose primary goal was not to assess depression. Hence, even though the results differed, the older adults in both groups had an overall GDS score within the normal range, which indicates that the results may not represent increased depression.

No statistically significant differences were observed between the groups in the comparative analysis of dance and various interventional approaches (i.e., meditation, weightlifting, aerobic exercises, physician counseling, and others). This contrasts the findings in the comparison of dance with noninterventional comparators. Among the seven studies that employed different activities as controls [41,43,44,46,50,51,55], six consistently demonstrated a reduction in depression scores postintervention, not only after dancing but regardless of the alternative intervention administered. In one study, lower depression scores were reported in the control group, who received physician counseling alone, compared to the dance group, who received both dance classes and physician counseling [55]. It is important to emphasize that most of the activities that led to improved scores involved group participation [41,43,44,46,51], fostering socialization and interaction among participants. Consequently, the improvement in depressive scores may not be solely attributed to dance itself but rather to the underlying influence of socialization, interpersonal engagement, and the “care effect” arising from the sense of receiving support and attention [63,64].

An interesting finding from the subgroup meta-analysis was that participants with PD did not show a significant improvement in depression following dance interventions. Depression is a well-known and prevalent nonmotor symptom of PD that is often challenging to manage in this population and sometimes exacerbated by standard dopaminergic drugs used for symptomatic control [65]. Of the four studies involving this population, two independently showed significant results favoring dance as a therapeutic approach [50,53], while two did not. However, it is important to note that in Michels et al., the control group participated in educational group activities, which could have influenced intergroup analysis, as all participants, irrespective of the intervention, had social interactions with others [51]. Additionally, in Rios et al., few participants exhibited baseline depression, possibly reducing the study’s statistical power [52]. Nevertheless, comorbidities coexisting with anxiety and depression should be considered, as there can be a mutual influence between these conditions.

Concerning anxiety, it was consistently reduced across all included studies that analyzed this outcome [16,40,42,43], except for one [41]. These studies involved dance sessions lasting, on average, 55 min, conducted three times a week for six weeks. Participants exhibited no other neurological or psychiatric comorbidities and presented high anxiety levels before the interventions. Therefore, the consistent trend in the results may be attributed to the more homogeneous baseline characteristics of the individuals and the smaller number of included studies, distinct from those that analyzed depression. Considering these characteristics and the suspected presence of publication bias, definitive conclusions for this outcome could not be ascertained. In contrast to our meta-analysis results, some studies concluded that dance has the potential to reduce anxiety in older adults with dementia [66]. Furthermore, various forms of physical activity, such as resistance exercises, aerobic exercises, sports participation, and others, have demonstrated positive impacts on anxiety [67].

The majority of the included studies were conducted in China [43,47,54,55], the USA [49,51,56], Brazil [42,45,46], and Spain [40,41]. Consequently, the publications originate from various continents, excluding Australia and Africa. Interestingly, these continents are associated with the lowest reported aggregate prevalence of depression [68]. Notably, Africa contributes the smallest fraction (<3%) of scientific publications among all continents [69]. Despite these regional disparities, a noteworthy global interest exists in seeking alternative/complementary treatments for depression and anxiety in older adults. The conduction of studies in other countries/continents would be crucial to provide more widely applicable and generalizable data.

An important aspect to discuss is the diversity of study methods employed across the publications, including the duration and schedule of the dance interventions. Evidence indicates that, for physical health, there is a positive relationship between the duration of exercises and the improvement of mortality scores [70]. However, for mental health purposes, such a relationship remains unclear. A recent umbrella review has shown that shorter-duration exercises (less than 12 weeks) more significantly improved mood scores compared to longer interventions, likely due to higher compliance to interventions in shorter periods [71]. On the other hand, another systematic review showed that intervention lasting more than 20 weeks had a greater potential to improve both cardiovascular and mental health outcomes [72]. Regarding exercise schedules, publications have demonstrated that engaging in moderate-intensity activities for more than 150 min every week could lower the risk of depression [73], which interestingly aligns with the average time of weekly interventions in the included studies for both depression and anxiety.

The prevalence of women in the included studies is a noteworthy aspect warranting discussion. This pattern may be attributed to historical cultural norms favoring dance as a form of expressive outlet for women [74]. Furthermore, the higher likelihood of women seeking mental health support and actively participating in therapeutic activities might contribute to their over-representation [75].

Our findings align with those of previously published systematic reviews on related topics, particularly in the context of depression [23,24,76,77,78]. However, none of these studies specifically aimed to discern whether the amelioration in depressive/anxiety symptoms could be attributed to dancing interventions or other influencing factors. In this regard, our study delves deeper into these aspects, revealing that underlying factors such as socialization and the care effect significantly contribute to these outcomes.

This study has some limitations. Due to the considerable heterogeneity presented in the included studies, we were unable to conduct a specific investigation into the extent to which distinct dance styles or schedules might have a more pronounced impact on reducing depressive and anxiety symptoms. Similarly, the feasibility of conducting subgroup analysis was limited, primarily owing to the small number of studies that addressed each condition (i.e., MCI and PD) and the relatively small sample sizes in certain studies. Another limitation was the diversity of assessment tools used to measure anxiety and depression symptoms within the included studies. Additionally, a restricted number of studies provided information on whether study participants were using antidepressant and anxiety medications, constituting a factor that could potentially influence the study outcomes. Future research should explore diverse dance styles and schedules within the studies, incorporating larger participant sample sizes to increase statistical power and enable more comprehensive assessments. Furthermore, a need exists for studies to employ more standardized assessment tools to evaluate depression and anxiety in diverse settings, including but not limited to community settings, hospitals, and nursing homes.

Our findings underscore the potential clinical relevance of integrating dance into mental health interventions for older adults, thereby highlighting a promising avenue for enhancing the mental well-being of this population.

## 5. Conclusions

Dance interventions demonstrated a reduction in depressive symptoms among older adults, albeit with a moderate risk of bias and very low certainty in the evidence. Notably, dance interventions did not exhibit superior effectiveness compared to other intervention types in control groups, a substantial proportion of which entailed group interactions. Depression scores were also not significantly different between older individuals with MCI and control groups. It was not possible to draw conclusions regarding the effect of dance intervention in studies that assessed anxiety, primarily due to the limited number of included studies and the presence of suspected publication bias.

## Figures and Tables

**Figure 1 behavsci-14-00043-f001:**
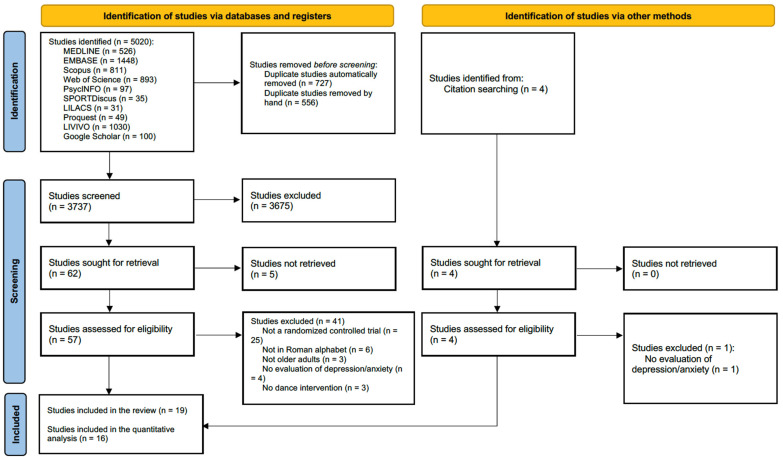
Flow diagram for study selection.

**Figure 2 behavsci-14-00043-f002:**
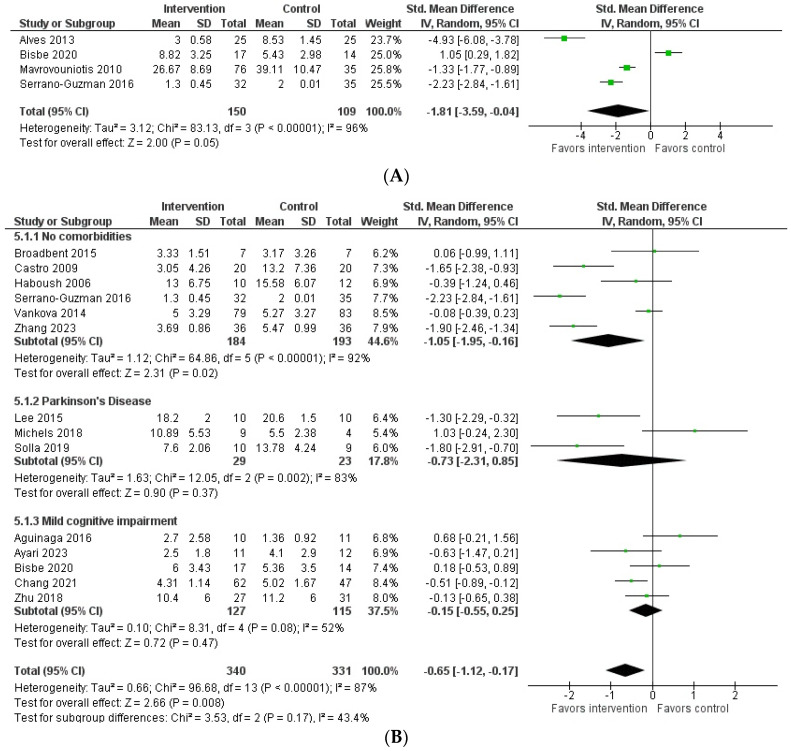
(**A**) Forest plot comparing anxiety symptoms in older adults between intervention (dance) groups and control groups (no interventions and other interventions). CI = confidence interval; df = degrees of freedom; IV, random = inverse variance, random-effects model; SE = standard error; SMD = standardized mean difference [16,40,41,42]. (**B**) Forest plot comparing depression symptoms in older adults between intervention (dance) groups and control groups (no interventions and other interventions). Subgroups were separated according to the study participants’ comorbidities. CI = confidence interval; df = degrees of freedom; IV, random = inverse variance, random effects model; SE = standard error; SMD = standardized mean difference [17,40,41,44,45,46,47,49,50,51,53,54,55,56]. (**C**) Forest plot comparing depression symptoms in older adults between intervention (dance) groups and control (no intervention) groups. Subgroups were separated according to the study participants’ comorbidities. CI = confidence interval; df = degrees of freedom; IV, random = inverse variance, random effects model; SE = standard error; SMD = standardized mean difference [17,40,45,46,47,49,54,56]. (**D**) Forest plot comparing depression symptoms in older adults between intervention (dance) groups and other interventions. Subgroups were separated according to the study participants’ comorbidities. AE = aerobic exercises; CI = confidence interval; df = degrees of freedom; IV, random = inverse variance, random effects model; M = meditation; NDT = neurodevelopment treatment; PC, physician counseling; TT = talk therapy; SE = standard error; SMD = standardized mean difference; WL = weightlifting exercises [41,44,46,50,51,55].

**Figure 3 behavsci-14-00043-f003:**
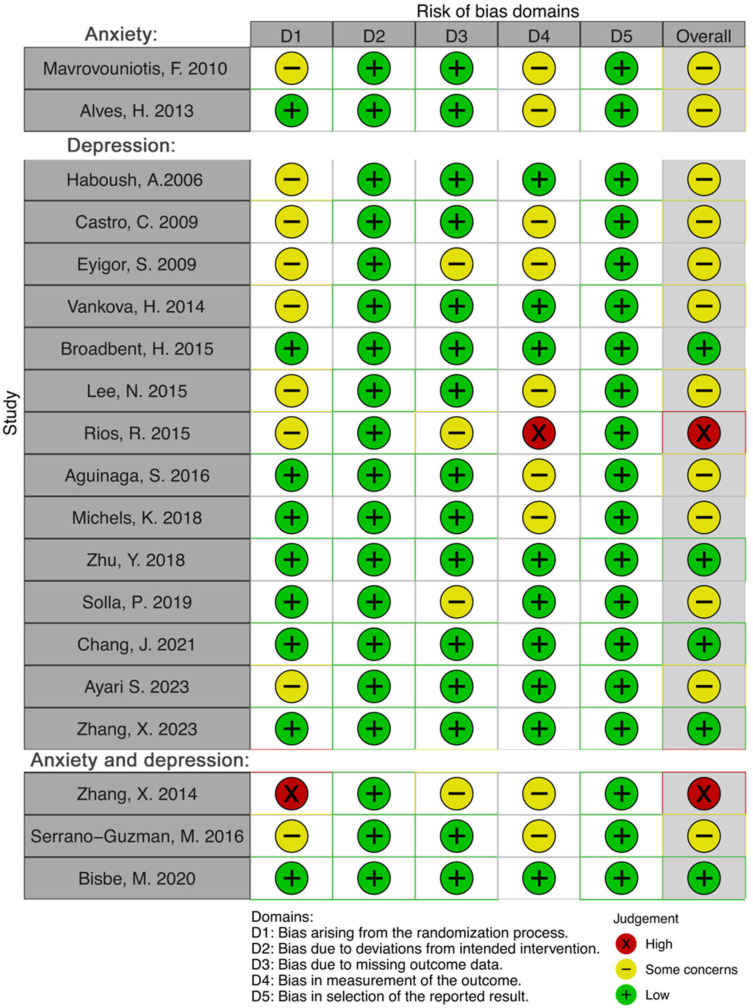
Risk of bias assessment for the included studies using the Cochrane risk-of-bias tool for randomized trials [16,17,40,41,42,43,44,45,46,47,48,49,50,51,52,53,54,55,56].

**Figure 4 behavsci-14-00043-f004:**
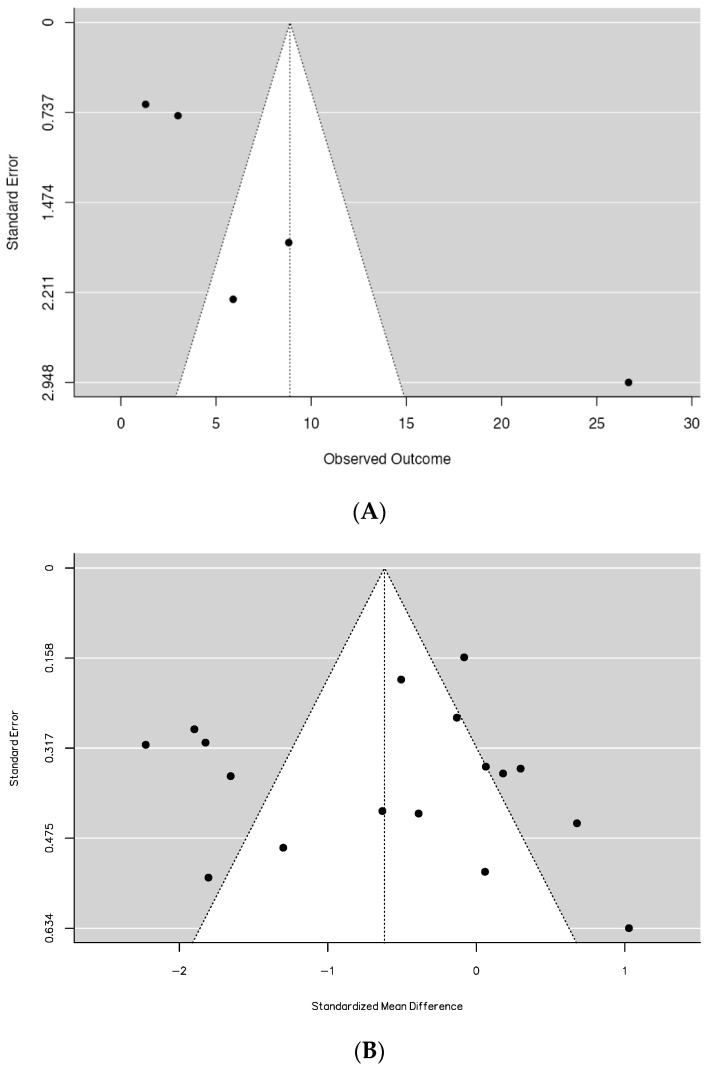
(**A**) Funnel plot assessing publication bias for studies on anxiety; (**B**) funnel plot assessing publication bias for studies on depression.

## Data Availability

The raw data supporting the conclusions of this article will be made available by the authors on request.

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
