# Peer review of "Effect of Dancing Interventions on Depression and Anxiety Symptoms in Older Adults: A Systematic Review and Meta-Analysis"

_behavsci, 2024, doi:10.3390/bs14010043_

Round 1
Reviewer 1 Report
Comments and Suggestions for Authors
I would like to thank you for the opportunity to review this article. This article denotes a great effort by the authors to perform it. The article covers a very interesting and current topic. Nevertheless, in my opinion, some parts need to be improved. I detail the comments below:
TITLE
- P. 1 line 1. If any conclusions about anxiety could not be drawn, maybe you have to rewrite the tittle in order to do it more adequate.
ABSTRACT
- P. 1 line 24. You can also focus on the clinical application.
KEYWORDS
- P. 1 line 25. Maybe “psychiatric disorders” is not a good keyword for this article. Probably Parkinson, or cognitive impairment could be more appropriate.
INTRODUCTION
- P. 1, line 42. Please, could you add a reference to the specified definition about Dance?
- P. 2, line 50. In my opinion, more information on the advantages of dance as a therapeutic exercise should be included. In addition, the benefits of dance as a leisure activity for physical, cognitive and social well-being can also be included.
- P. 2, line 56. It could be interesting to include more information about the prevalence or socio-demographic data of the psycho-emotional disruption in the Parkinson subjects or in the cognitive impairment.
- P 2, line 62. Do you know if previous authors have demanded more investigation with a higher strictness about this topic? If so, could you specify it?
METHODS
- P 2, line 78. Can you reference the definition of older adults as subject with an age of older than 60 years?
- P 2, line 80. The eligibility criteria included, you have mentioned “comparing the data with a control group in which no dance intervention”, so probably you should check the C (Comparator) of the PICOS.
RESULTS
- P 6, line 240. Could you include the I2 value of the anxiety M-A.
- P 6, line 243. It could be interesting to performed a sub-analysis in the meta-analysis comparing dancing intervention versus the other physical activity interventions or therapies, and dancing versus no intervention or usual care.
- P 6, line 244. Errata “indicatd”.
- P 8, line 324. Errata, the results of the certainty of evidence is in Table S2 not S1.
- P 9, table 1. In discussion section you have talked about the importance of developing the intervention on hospital or nursing homes, so it could be interesting to include in the table 1 other information as the setting of the intervention, or the evaluated outcomes.
- P 9, table 1. The article of Serrano-Guzman, 2016 (31) have not used the Euroqol-5D for assessing the anxiety or depression. Could you clarify me why it is presented in the table?
DISCUSSION
In my opinion, the discussion section is a bit long, it is necessary to be a bit more specific about the provided insights. In addition, in this section it is important to provide more comparisons with previous studies to show the new findings or reinforcements provided by this article.
It could be also interesting to discuss why the most part of the sample are women.
You could add a section to talk about the strength and the clinical application of your study.
REFERENCES
- P 16, line 561. Some reference format need to be check, for example, references n28, or 30.
Reviewer 2 Report
Comments and Suggestions for Authors
Dear Editors and authors,
Thank you for inviting me to review this manuscript. First of all, I would like to congratulate the authors on the great quality of their manuscript. The present review and meta-analysis is a relevant work that helps to understand the effect and potential of dancing interventions on mental health. However, there are some opportunities for improvement that I hope the authors consider.
Results
In Figure 2B instead of “Control”, the heading of the third column should not be “other interventions”.
Please increase the size of the figures 2A, 2B, 2C y 2D.
3.3.1 Depression
Line 243: The heading of "Depression" should be "3.3.2 Depression".
Line 244: "indicatd" needs to be corrected.
3.4. Risk of bias
Lines 311-315: In this paragraph "one study" is repeated too much.
In table 1 instead of "Dance duration", "Dance program duration" may be more suitable.
Discussion
Line 446: Letter size of the phrase "health purposes, such a relationship remains unclear" may need to be reduced.
Comments on the Quality of English LanguageEnglish quality is good, however there is one spelling mistake in line 244.
Reviewer 3 Report
Comments and Suggestions for Authors
The authors have provided a well developed and written exploration into the effects of dance on depression and anxiety in older adults. The paper is presented very methodically and is well written.
The only constructive comment I would make is that I recommend adding information in section 3.2 Study Characteristics on the use of medications for anxiety and/or depression in the reviewed studies as these may impact reported outcomes. If medications were not described in the reviewed studies, this would be an additional limitation.
There were two spelling errors that I caught:
Line 99 - evaluating
Line 144 - indicated
The authors should be commended for submitting this near publication ready manuscript. Thank you for your diligent work.
